# Addressing Antiretroviral Drug Resistance with Host-Targeting Drugs—First Steps towards Developing a Host-Targeting HIV-1 Assembly Inhibitor

**DOI:** 10.3390/v13030451

**Published:** 2021-03-10

**Authors:** Jaisri R. Lingappa, Vishwanath R. Lingappa, Jonathan C. Reed

**Affiliations:** 1Department of Global Health, University of Washington, Seattle, WA 98109, USA; reedjon@uw.edu; 2Prosetta Biosciences, 670 5th Street, San Francisco, CA 94107, USA; vlingappa@prosetta.com

**Keywords:** HIV-1 assembly, HIV-1 capsid, Gag, antiretroviral, antiviral, drug screen, viral-host interactions, ABCE1, DDX6, RNA granule

## Abstract

The concerning increase in HIV-1 resistance argues for prioritizing the development of host-targeting antiviral drugs because such drugs can offer high genetic barriers to the selection of drug-resistant viral variants. Targeting host proteins could also yield drugs that act on viral life cycle events that have proven elusive to inhibition, such as intracellular events of HIV-1 immature capsid assembly. Here, we review small molecule inhibitors identified primarily through HIV-1 self-assembly screens and describe how all act either narrowly post-entry or broadly on early and late events of the HIV-1 life cycle. We propose that a different screening approach could identify compounds that specifically inhibit HIV-1 Gag assembly, as was observed when a potent rabies virus inhibitor was identified using a host-catalyzed rabies assembly screen. As an example of this possibility, we discuss an antiretroviral small molecule recently identified using a screen that recapitulates the host-catalyzed HIV-1 capsid assembly pathway. This chemotype potently blocks HIV-1 replication in T cells by specifically inhibiting immature HIV-1 capsid assembly but fails to select for resistant viral variants over 37 passages, suggesting a host protein target. Development of such small molecules could yield novel host-targeting antiretroviral drugs and provide insight into chronic diseases resulting from dysregulation of host machinery targeted by these drugs.

## 1. Host-Targeting Antiviral Drugs—A Way to Fill a Gap in the Antiretroviral Armamentarium and Reduce the Specter of Antiviral Drug Resistance?

Currently, over 25 million of the 38 million people living with AIDS worldwide are being treated with antiretroviral therapy (ART), with the result being a 60% reduction in AIDS-related deaths relative to the peak of the AIDS pandemic in 2004 [1]. However, resistance to approved anti-retroviral (ARV) drugs continues to grow, threatening the huge gains made through ART over the past three decades [2]. In low- and middle-income countries (LMICs), the failure of first-line ART to contain viral load is as high as ~20% [3], and in some studies in Sub-Saharan Africa half of patients who have failed first line tenofovir-containing regimens have resistance to all three drugs in the regimen [2,4,5]. Additionally, a substantial increase in the rate of pretreatment drug resistance has been observed in LMICs, with the prevalence of resistance to some ARV drugs in areas of Africa and Latin America approaching the World Health Organization’s 10% threshold for changing first line therapy [6]. To underscore the urgency of the situation, the growing wave of treatment failures affecting resource-limited countries is now being called a fourth AIDS epidemic, with the first three AIDS epidemics being the worldwide spread of the virus, the disease toll caused by the virus, and the problematic social, economic, cultural, and political response to the first two epidemics [7]. Needless to say, in this era of the coronavirus pandemic, warning signs in resource-poor settings must be heeded by those in resource-rich settings who might have once considered themselves protected by their socioeconomic status. Moreover, the prevalence of transmitted drug-resistant HIV-1 variants is higher in Europe and especially North America than in LMICs [8]. Thus, there is a clear need for developing new ARV drugs, including those that are less prone to resistance and can therefore end the cycle in which use of new drugs leads to resistant viral variants that ultimately reduce the utility of those drugs.

To date, 28 medicines have been FDA-approved for the treatment of HIV disease, of which 20 are ARV small molecules that act by seven different mechanisms to inhibit specific steps in the viral life cycle (the others are combination drugs, peptides, and monoclonal antibodies; [9]). Nineteen of these small molecules are direct-acting antiviral agents (DAAs)—these target viral proteins (specifically Reverse Transcriptase, Integrase, Envelope, and Protease). DAAs are the mainstay of ARV treatment and have proven highly effective especially when given in combination. However, because the error-prone HIV reverse transcriptase results in generation of viral variants, treatment with DAAs can lead to selection for variants that harbor resistance mutations if viral replication is not fully and continuously suppressed [10]. Currently approved DAAs exhibit a relatively low genetic barrier to resistance, with fewer than four resistance mutations in the HIV-1 target typically leading to drug failure [11]. Fortunately, drug-resistant variants tend to exhibit reduced viral fitness; moreover, cross resistance between different classes of DAAs is not observed because mutations alter only the protein interfaces relevant to one drug class, allowing other classes of DAAs to be used when resistance is detected. Additionally, the problem of drug resistance is mitigated by the use of combination ART (cART) chosen based on resistance testing, particularly when cART fully suppresses viral replication thereby preventing the generation of viral variants. Nevertheless, because problems with adherence to DAAs are common and the barrier to resistance of DAAs is low, resistance continues to arise despite the use of combination therapy.

Of the 20 ARV small molecules only one is a host-targeting antiviral agent (HTA)—maraviroc, which targets the entry coreceptor CCR5 (reviewed in [12]). While DAAs are more straightforward to develop, more progress towards developing HTAs might have been expected by now for three reasons. First, it has long been known that host factors are critical at nearly every step in the viral life cycle [13]. Second, in contrast to viral genes, which mutate rapidly in response to selective pressure, host factors are relatively immutable; moreover, the number of mutations that are tolerated in viral proteins that bind to host factors is limited. Thus, drugs that target host proteins are likely to provide a high genetic barrier to development of drug-resistant viral variants, ensuring an extended shelf life for such drugs [13]. Third, it has been over a decade since maraviroc, the only approved ARV small molecule that acts on a host factor, was initially shown to be safe and effective [14]. Moreover, this finding was confirmed in longer term follow-up studies and in both treatment naïve and treatment-experienced patients (reviewed in [12]), proving that host-targeting ARV drugs can be successful. It would seem that such a success would have opened the door to innovative strategies for identifying host-targeting antiretroviral compounds that could hold back the tide of ARV resistance, but that has not been the case.

Notably, while the high genetic barrier to resistance provides a strong argument in favor of development of HTAs, there are also potential disadvantages to HTAs. Most importantly, host targeting raises concerns of side effects since the targeted host protein could be important for host functions. However, depending on their exact mechanisms of action, host-targeting drugs can be very safe, for example if the host is able to use other proteins or pathways in place of the drug target, if the HTA binds to its host target in a manner that allows the target protein to be used by the host but not by the virus, or if the virus alters the target so that it cannot be used by the host until rescued by the drug and returned to its functional state. The latter mechanism has been reported for a host-targeting drug that inhibits multiple respiratory viruses in a recent bioRxiv preprint [15], although further study will be needed to validate this mechanism.

Examples of HTAs that have serious side effects and those that are very well tolerated are found among drugs that inhibit the chemokine co-receptors (CXCR4 and CCR5) required for HIV-1 entry into cells (reviewed in [12]). The CXCR4 antagonist plerixafor was not pursued as an ARV treatment because it was found to increase cardiac dysfunction [16], although interestingly plerixafor was later approved as a hematopoietic stem cell mobilizer (reviewed in [12]). In contrast, the CCR5 antagonist maraviroc was well tolerated [14,17], even in 5- and 10-year retrospective studies [18,19]. Maraviroc does have important limitations, including that it can only be given to patients who are infected with CCR5-tropic HIV-1 strains (which are common early in disease and in treatment naïve patients) and that treatment failure occurs in ~12% of patients for reasons that are not well understood, perhaps because of chemokine receptor redundancy or inefficacy of blockade in some individuals (reviewed in [12]). However, the rate of treatment failure due to development of CXCR4-tropic HIV-1 strains is very low following long term maraviroc treatment, as is development of maraviroc-resistant strains [19]. Additionally, maraviroc has unexpected beneficial side effects, such as reduced atherosclerotic progression [20,21]. As more HTAs are developed, it will be interesting to see whether beneficial effects on progression of other diseases are common features of HTAs. Regardless, use of maraviroc over more than a decade shows that HTAs can be safe and effective. Moreover, because of their high genetic barrier to resistance, HTAs would be expected to serve as excellent adjuncts to DAAs in combination regimens.

Host-targeting antiretroviral compounds would be of particular interest for stages of the life cycle that have not yet been successfully targeted. The HIV-1 life cycle can be broadly divided into the early events, which encompass initial infection of a target cell through proviral DNA integration, and late events, which encompass viral transcription through production of infectious virus particles (Figure 1). The 20 ARV drugs that are currently FDA-approved act at four different steps in the viral life cycle, three of which are early events—viral entry, reverse transcription, and integration. Only one of the steps targeted by FDA-approved ARV drugs occurs during viral late events—specifically, maturation, the last step of the HIV-1 late events, is blocked by the protease inhibitors, which are an important mainstay of combination antiretroviral therapy (cART). Thus, inhibitors have yet to be identified for most viral late events, including the events involved in assembly of the HIV-1 immature capsid (pink bar in Figure 1). This obvious gap in the viral armamentarium has generated great interest in identification of assembly inhibitors over the past two decades (reviewed in [22]).

While there are different models for how immature HIV-1 capsid assembly occurs in cells, the important roles of Gag-RNA interactions and Gag targeting to the plasma membrane (PM) are widely accepted. HIV-1 Gag assembly is initiated in the cytoplasm, when a dimer or oligomer of Gag associates with unspliced HIV-1 genomic RNA [23,24]. This Gag-RNA complex targets to the host cell plasma membrane (PM), where Gag multimerizes to form the spherical completed immature capsids, each of which is composed of thousands of newly synthesized HIV-1 Gag and GagPol polyproteins that package the HIV-1 RNA genome. Budding and envelopment of the assembled immature capsid at the PM results in release of immature HIV-1 particles (reviewed in [25]). Subsequently, during maturation, Gag polyproteins are cleaved by the HIV-1 protease (encoded in the *pol* gene) to yield four component proteins—matrix (MA), capsid (CA), nucleocapsid (NC) and p6 (reviewed in [26]). Rearrangement of the cleaved Gag proteins yields fully infectious viral particles that contain the mature capsid, also called the core, a cone-shaped structure composed of CA (Figure 1). An important property of the mature core is its metastability—the core needs to be stable enough to protect the viral genome after egress but labile enough to disassemble at the optimal time post-entry for releasing the viral genome into the cytoplasm. This delicate balance is disrupted by mutations in CA that that tip the balance in either direction—excessive stability or premature disassembly—leading to a reduction in productive infection (reviewed in [27]).

Given the underlying molecular biology, one might expect HIV-1 immature capsid assembly to be a particularly good target for inhibition by small molecules. Structures composed of multimers, like capsids, should be highly susceptible to inhibition, since an inhibitor need bind only a few of the thousands of Gag subunits in the immature capsid to disrupt capsid formation and/or function [28]. Consistent with that prediction, inhibitors that result in non-infectious virus have been successfully identified—these compounds bind to CA leading to virus that is unable to complete maturation and the post-entry events of uncoating and/or reverse transcription (reviewed in [22]). However, one would also expect the identification of potent compounds that *specifically* inhibit assembly of immature capsids, leading to reduced virus production, but such compounds had not been identified as of 2019. One reason for the lack of potent and specific inhibitors of immature capsid assembly could be that assembly is able to proceed when a drug is bound to a few Gag subunits, while the later maturation and post-entry events are less tolerant of drug binding. Alternatively, it is possible that screens have missed promising small molecule inhibitors of immature capsid assembly, perhaps because the screens that were used did not recapitulate events of immature capsid assembly and focused instead either too narrowly on one aspect of assembly or too broadly. Support for this latter hypothesis is provided by the identification by our group in 2020 of a potent and specific inhibitor of HIV-1 immature capsid assembly using a novel screening approach that reconstituted the entire pathway of HIV-1 immature capsid assembly, including both Gag-Gag interactions and Gag-host interactions [29], as discussed in more detail below.

## 2. Spontaneous Assembly or Host-Catalyzed Assembly of HIV-1 Gag? Two Models with Implications for Assembly Inhibitors

The working model one uses to study a stage of the viral life cycle influences the design of drugs screens used to identify inhibitors, which in turn results in some inhibitors being identified while other promising inhibitors are missed—thus the starting model matters. For decades the dominant model for understanding Gag multimerization has been the self-assembly model, which proposes that Gag polypeptides multimerize spontaneously in the presence of nucleic acids due to intrinsic properties that promote Gag-RNA and Gag-Gag interactions (reviewed in [30,31,32]). This model has been supported by in vitro studies in which assembly of recombinant Gag peptides is studied in the presence of nucleic acids but in the absence of the host proteins that are present when Gag assembles in cells. These studies have revealed important properties of Gag domains in promoting Gag-Gag and Gag-RNA interactions. They have also evolved over time to incorporate a role for host phospholipids such as IP6 in self-assembly [31]. However, evidence, described below, that intracellular Gag associates with viral and cellular RNA by trafficking to host ribonucleoprotein (RNP) complexes has not been incorporated into the self-assembly model [31,33,34].

Data in support of host-catalyzed Gag assembly was first generated when, in an attempt to better approximate HIV-1 infection in vivo, assembly of nascent Gag polypeptides was studied in the context of host proteins, using cell extracts (a cell-free system). Unlike self-assembly studies, which examine recombinant Gag at high concentrations in the complete absence of host proteins [35,36], the cell-free system allowed a direct test of the hypothesis that Gag assembles spontaneously in a cellular environment. In the cell-free system, where newly synthesized Gag is present at concentrations more typical of what is observed in infected cells, Gag was found to assemble into particles that closely resembled immature capsids in an energy-dependent manner—not spontaneously; moreover, the energy dependence was post-translational and was therefore independent of the energy requirement expected during Gag synthesis [37,38]. Pulse-chase studies revealed that during assembly in the cell-free system as well as in cells, Gag progresses through a pathway of assembly intermediates that are transient, sequential multiprotein complexes of different sizes, defined by their approximate sedimentation (S) values (~10S, ~80S/150S, and ~500S) [37,39]. Progression of Gag along this pathway culminates in formation of the ~750S completely assembled immature capsid. The order of assembly intermediates revealed by pulse-chase studies [37,39] was confirmed by analysis of assembly-defective Gag mutants, each of which was found to be arrested at a characteristic step in the pathway ([37,39,40,41,42]; Figure 2). ATP hydrolysis is also required at a discrete point in the assembly pathway ([37]; Figure 2). Since HIV-1 Gag does not bind and hydrolyze ATP, these findings indicated that in a cellular context, Gag assembly is energy-dependent and catalyzed by host enzymes. Consistent with this observation, two host ATPases were subsequently identified as facilitators of HIV-1 Gag assembly. The first was ATP-Binding Cassette protein E1 (ABCE1), an essential and highly conserved enzyme found in eukaryotes and archaebacteria. Knockout of ABCE1 leads to rapid cell death [43] most likely because it is required for recycling of ribosomes (reviewed in [44]); thus, the role of ABCE1 in promoting post-translational steps in immature capsid assembly was demonstrated using depletion and reconstitution experiments in cell extracts and dominant negative experiments in cells [45]. Subsequently, a second ATPase, the Dead-box RNA helicase DDX6, was also shown to facilitate HIV-1 immature capsid assembly in cells, with loss of assembly upon knockdown, and rescue of assembly by wild-type DDX6 but not by an ATPase deficient DDX6 mutant [46].

**Figure 2 viruses-13-00451-f002:**
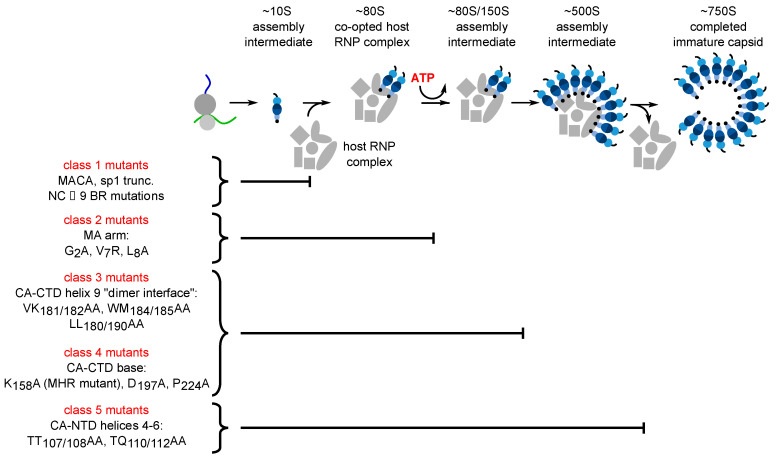
Each assembly-defective Gag mutant is arrested at a characteristic point in the HIV-1 capsid assembly pathway. Initial studies in the cell-free system and in HIV-1-expressing cells revealed that Gag progresses through a stepwise pathway of multiprotein complexes defined as intermediates in a pathway of assembly based on pulse-chase experiments (reviewed in [47])). Subsequent experiments confirmed the order of intermediates in this pathway by showing that each assembly-defective Gag mutant is arrested at a characteristic point in this pathway. Altogether five different categories of mutants have been identified (Class 1–5), one for each point in the pathway. The difference between Class 3 and Class 4 being that Class 3 mutants are arrested as cytosolic ~80S complexes while Class 4 mutants are arrested as PM-associated ~80S complexes, suggesting that the ~80S assembly intermediate is the complex in which assembling Gag traffics from the cytosol to the PM. Later studies identified some of the host proteins in the assembly intermediates. Relevant references are in the text.

The mechanisms by which ABCE1 and DDX6 promote Gag assembly have not been fully elucidated. However, in the case of DDX6, clues come from the known role of RNA helicases in RNP remodeling, a process by which proteins that are associated with RNAs and dictate RNA fates are replaced by other proteins [48]—thus, DDX6 could facilitate the association of Gag with viral RNA during assembly. In keeping with this possibility, in HIV-1 expressing cells DDX6 is known to colocalize with HIV-1 genomic RNA at PM sites of virus assembly and budding [49]. HIV-1 Gag is also colocalized with DDX6 and ABCE1 in situ, including at the PM at sites of budding [29,39,41,46,49,50]. Together the data support a model in which, in cells, newly synthesized Gag traffics to—and assembles in—specific host RNP complexes that contain both the HIV-1 genomic RNA that is packaged into the assembling virus and host enzymes that facilitate the assembly and packaging process (Figure 3A). Additionally, by sequestering the assembly process within host RNP complexes where non-translating cellular RNAs are stored, the virus is likely able to shield assembling progeny virus from host innate immune effectors.

Despite fundamental differences between the host-catalyzed and spontaneous models of Gag assembly, there are many ways in which the host-catalyzed model complements the self-assembly model. For example, the self-assembly studies show that RNA plays a key role in promoting Gag-Gag interactions [32]. However, unlike self-assembly reactions that contain recombinant Gag and purified RNA, the cytoplasm lacks soluble RNA, with all host and viral RNAs being instead sequestered within multiprotein complexes containing host RNPs [49]. Indeed, the naked viral RNA molecules shown in many viral life cycle diagrams are problematic given the lack of evidence for naked viral RNA genomes in the cytoplasm of infected cells. Data in support of the host-catalyzed model demonstrate that Gag associates with RNA in cells by trafficking to host RNP complexes, thereby forming assembly intermediates that contain Gag, HIV-1 genomic RNA, ABCE1, DDX6, and other host proteins [41,49]. Thus, assembly intermediates appear to be formed when Gag co-opts host RNP complexes that exist in uninfected host cells [49], setting the stage for Gag to replace host RNPs associated with HIV-1 genomic RNA. The RNP complexes that are co-opted resemble host RNA granules since they can be detected by light microscopy but are distinct from the much larger and better studied DDX6-containing RNA granules called P-bodies [29,49]. Like canonical RNA granules, the co-opted host RNP complexes are likely sites of host mRNA storage and metabolism, repurposed by the virus for the purpose of immature capsid assembly and genomic RNA packaging.

Similarly, data showing that HIV-1 utilizes host ATPases to facilitate virus assembly [45,46], thereby increasing the efficiency of virus progeny production, make sense from a broader virologic perspective given that viruses with larger genomes such as DNA viruses encode their own ATPases that function during packaging and assembly [51]. Thus, the host-catalyzed model of Gag assembly is in keeping with basic principles of virology and RNA cell biology.

In a recent paper Deng et al. argue against the existence of the stepwise pathway for HIV-1 assembly [52], but for multiple reasons this study is problematic. First, Deng et al. conclude that all Gag-containing complexes that comigrate with ribosomes likely consist of Gag-binding to ribosomes based solely on their finding that recombinant Gag-GFP purified from bacteria can bind to purified ribosomes, presumably nonspecifically, and without showing that Gag binds to ribosomes in cells by a similar mechanism [52]. They did not address the possibility that complexes containing Gag bound to ribosomes, if they exist in cells, could co-exist with similar sized Gag-containing complexes of an entirely different composition such as assembly intermediates. Notably, well established techniques allow assembly intermediates to be isolated away from other Gag-containing complexes of similar sizes, including ribosomes, by immunoprecipitation using antibodies directed against ABCE1 and RNA granule proteins, the host proteins that are components of these assembly intermediates [39,41,45,46,49,50,53,54]. As Deng et al. did not use such techniques to isolate assembly intermediates, it is likely that they were studying other complexes in addition to assembly intermediates, such as translating Gag associated with ribosomes, newly synthesized Gag, Gag undergoing degradation, and Gag in complexes that have partially dissociated. Indeed, the Gag-containing ribosomal complexes described by Deng et al. [52] are almost certainly not assembly intermediates, since assembly intermediates contain proteins that are not found in ribosomes, such as the RNA granule proteins DDX6, AGO2, and DCP2, and lack the abundant small ribosomal protein S6 [29,41,46,49,50,54]. Notably, complexes containing Gag and RNA granule proteins are not biochemical artifacts since they have also been found in situ using two different imaging techniques in multiple cell types [29,41,46,49,50,54].

A second reason why Deng et al. may have failed to detect assembly intermediates relates to their use of antibody-mediated detection of Gag epitopes to assess the multimerization state of Gag-containing complexes. This technique is problematic since it is well known that Gag epitopes become inaccessible as Gag multimerizes [55]. Given that completed immature capsids contain thousands of Gag proteins [56], the detection of mainly Gag monomers and dimers by Deng et al. (Figure 3 in [52]) raises the possibility that their approach simply excluded late assembly intermediates. Adding to that concern, no positive control was provided to demonstrate that their method is capable of capturing higher order Gag multimers. In a third problematic approach, Deng et al. argue that the smaller Gag-containing complexes found in cells do not behave like assembly intermediates since they are labeled equivalently during SILAC pulse labeling [52]. However, given that assembly happens very rapidly (in <15 min based on imaging in [23,57]), a pulse label of 2 h, the shortest pulse used by Deng et al. [52], would be expected to label many cohorts of assembling Gag and would actually approximate steady-state labeling which lacks the resolution needed to dissect the kinetics of a highly transient process. In contrast, the pulse-chase conditions that originally identified the HIV-1 assembly intermediates involved a much shorter pulse of radiolabel (4–15 min) followed by unlabeled chase periods of varying lengths, allowing small cohorts of assembling Gag to be followed as the size of these Gag-containing complexes changed over time and thereby establishing the progression of a small population of labeled Gag through complexes of increasing size [37,39]. Additionally, for pulse chase experiments performed in HIV-1-expressing cells, anti-ABCE1 immunoprecipitation was used to isolate Gag-containing assembly intermediates from other Gag-containing complexes [37,39]. At later times, much of the radiolabeled Gag that was observed initially in ~80S and later in ~500S ABCE1-containing complexes was no longer in those complexes, appearing instead in released virus like particles (VLPs) in the medium, suggesting that the ~80S and later the ~500S ABCE1-containing complexes are precursors to released VLPs (Figures 2 and 5 in [37,39]).

Finally, because Deng et al. find that the assembly-defective G2A Gag mutant forms a complex that is similar in size to our ~500S assembly intermediate, they argue that the ~500S complex formed by WT Gag must not be a late assembly intermediate. However, multiple groups have shown that non-myristoylated G2A Gag can overcome the assembly block, multimerize, and produce virus like particles in the cytoplasm [58,59,60], most likely due to overexpression. In the absence of controls for overexpression [52], Deng et al. cannot rule out that in their hands G2A multimerized to form late assembly intermediates and VLPs, with the VLPs being cytoplasmic as observed by others [58,59,60] and therefore not forming the puncta characteristic of membrane-associated assembly.

Thus, while many aspects of the assembly pathway we have proposed remain to be understood, the study by Deng et al. did not employ approaches that would have allowed them to make conclusions about HIV-1 assembly intermediates and therefore sheds little light on the stepwise pathway of HIV-1 capsid assembly. Those concerns having been noted, probably the best approach to validating any viral life cycle model is to use that model to identify novel classes of inhibitors that are predicted by the model and can also be used to further understand the model.

## 3. Drugs Screens Over Two Decades Have Identified Small Molecule Inhibitors That Bind to CA

Until now, two screening approaches have been used to identify small molecules that inhibit viral late events and/or bind to CA: (1) hypothesis-neutral cell-based replication assays; and (2) in vitro reactions that assay self-assembly of recombinant CA peptides or CA-NC peptides (the latter encompasses CA and the adjacent NC domain of Gag). As HIV-1 capsid inhibitors have been reviewed relatively recently [22], here we will describe some of the best studied small molecule inhibitors that have emerged from these two types of assembly screens.

A full viral replication screen yielded the CA-binding small molecule PF74 (PF-3450074; [61]). PF74 displays sub-micromolar potency and inhibits early post-entry events in the viral life cycle as well as integration and viral late events [62,63,64,65]. PF74 binds a conserved pocket in CA-NTD at the interface with CA-CTD that also serves as the binding site for the host factors cleavage and polyadenylation specific factor 6 (CPSF6) and nucleoporin 153 (NUP153); thus, PF74 may act as a competitive inhibitor of CPSF6 and NUP153 binding to CA [62,66,67]. Consistent with crystal structure data, resistance mutations that arise in CA upon selection with PF74 are located in the region of the PF74 binding site [68].

Most other screens for assembly inhibitors have involved self-assembly assays, which largely focus on self-assembly of recombinant CA, in part because of the critical role played by CA during immature and mature capsid assembly as well as during capsid disassembly. The CA domain of the Gag polyprotein contains residues critical for Gag-Gag interactions that mediate multimerization during assembly of the immature capsid. Subsequently, after Gag cleavage, CA is a key component in assembly of the mature capsid. After target cell entry, CA disassembly is required for reverse transcription, trafficking of the pre-integration complex to the nucleus, and integration (reviewed in [69]). CA consists of two subdomains, the N- and C-terminal domains of CA (CA-NTD and CA-CTD), whose crystal structures have been solved [70], allowing the self-assembly and replication assay screens described above to be complemented by in silico modeling.

A CA self-assembly assay was used to validate the first CA-targeting small molecule, CAP-1, which was initially identified in silico as a potential CA-binding compound [71]. CAP-1 results in production of non-infectious virions with abnormal cores [71] and binds to an induced hydrophobic pocket at the base of CA-NTD that is distinct from the PF74 binding pocket [71,72]. After identification of CAP-1, a high throughput CA-NC peptide self-assembly screen of a compound library identified two additional groups of small molecule inhibitors: benzimidazole (BM) compounds, which act in a manner similar to CAP-1 to produce non-infectious virus; and benzodiazepine (BD) compounds, which allow Gag processing but not release suggesting a budding defect [73]. BM and BD compounds bind to the same pocket in CA-NTD as CAP-1 but with expanded contacts, as shown in crystal structures and confirmed by selection in cell culture yielding resistance mutations in the region of the binding site in CA ([73]. While CAP-1 inhibits HIV-1 replication with a half maximal effective concentration (EC50) in the micromolar range [71], the best BD and BM compounds are more potent, with EC50s of 60–70 nM and a large difference between the 50% cytotoxic concentration (CC50) and the EC50 (CC50/EC50 > 300; [73]).

Other CA-binding small molecules have also been identified using variations of the self-assembly assay. For example, ebselen, an organoselenium compound, was identified using a high throughput screen that monitored recombinant CA-CTD peptide dimerization (the first step of self-assembly) using time-resolved fluorescence resonance energy transfer [74]. Ebselen binds to CA-CTD and inhibits viral replication at an early, post-entry stage with an EC50 of 3.2 µM and a CC50 of >30 µM in peripheral blood mononuclear cells (PBMCs) [74].

More recently, self-assembly screens yielded the most promising CA-binding small molecule identified to date, GS-6207 [75]. Both GS-6207 (GS-CA2) and the related small molecule GS-CA1 bind to the pocket at the interface of CA-NTD and CA-CTD that also binds PF74, CPSF6, and NUP153 [76,77]. In keeping with this, selection results in resistance mutations that map to this binding site. GS-6207 and GS-CA1 inhibit HIV-1 replication with EC50s of 60 pM and 32 pM, respectively, in human CD4+ primary T cells, and CC50 values of >50 µM [76,77]. Both GS-6207 and GS-CA1inhibit at multiple points in the early and late parts of the viral life cycle, as is the case for PF74, but with greater potency against early events. Thus, because of this broad targeting of early and late events, GS-6207 and GS-CA1 lack specificity. However, due in part to their very high potency, both compounds are amenable to long-acting injectable therapy [22,75], and GS-6207 has advanced to Phase 1 clinical trials, where it was found to be safe and well tolerated [76].

## 4. Screening for Host-Targeting Antiretroviral Compounds Using the Approach That Identified the First Rabies Virus Inhibitor

While numerous experimental approaches have validated the host-catalyzed assembly model in cells, aspects of the model have been difficult to completely prove using currently available approaches. For example, since the assembly intermediates are highly transient and labile multiprotein complexes that are present in minute quantities, purifying them and cataloguing their components by mass spectrometry has been challenging. Likewise, the host components in these complexes have not been visualized during live imaging of virus assembly in cells since tagging multiple host proteins with fluorophores is much more difficult and problematic than tagging multiple viral proteins. Similarly, knockdowns that produce log-fold effects are unlikely when numerous host enzymes are involved in facilitating assembly, each contributing only modestly and likely with redundancy. However, ultimately the goal of understanding steps in the viral life cycle is to find novel approaches to antiviral therapeutics; thus, one way to bypass the technical issues that have limited acceptance of the host-catalyzed assembly model is to simply ask if screens based on this model yield successful assembly inhibitors.

The first evidence that establishing screens based on host-catalyzed capsid assembly could be effective came from a study that identified the first potent small molecule inhibitor of rabies virus (RABV) replication in cell culture [78]. In this study, a putative pathway of RABV assembly intermediates was identified using a cell-free system analogous to that used to identify the HIV-1 immature capsid assembly pathway. This putative assembly pathway was then developed into a moderately high throughput plate screen, in which tagged antibodies to RABV capsid proteins emit a signal proportional to the degree of capsid protein multimerization, thus allowing small molecule inhibitors to be identified by their ability to inhibit this signal. This led to a RABV assembly inhibitor chemotype that was further advanced, yielding a potent small molecule analog that inhibits infectious RABV in Vero cell culture with a 50% effective concentration (EC50) of 15–30 nM and a 50% cytotoxic concentration (CC50) of 2.5–10 μM. Notably, when conjugated to a resin, the small molecule inhibitor pulled down a set of proteins that included ABCE1 [78], the host enzyme previously found to promote HIV-1 capsid assembly [45]. Moreover, the resin-bound proteins were required for inhibition by the small molecule in the plate-assay, suggesting that these proteins are the target of the inhibitor [78]. Thus, this study revealed that potent host-targeting small molecule inhibitors of virus replication can be identified through screens based on host-catalyzed viral capsid assembly pathways and raised the possibility that host machinery co-opted by different viruses may have similarities, as suggested by a common host protein component.

## 5. Identification of PAV-206, a Potent Small Molecule Inhibitor of HIV-1 Immature Capsid Assembly

The cell-free system in which the host-catalyzed HIV-1 assembly pathway was first identified was also used to develop an HIV-1 capsid assembly plate screen [29], analogous to that developed for RABV [78]. In this HIV-1 capsid assembly screen (Figure 3B), a cell extract is programmed with Gag mRNA, leading to synthesis of Gag polypeptides. These Gag polypeptides co-opt host RNP complexes that are present in the extract and contain the previously identified host facilitators of assembly, resulting in formation of sequential assembly intermediates. Plates containing wells coated with anti-Gag antibody result in capture of these Gag-containing assembly intermediates. A soluble anti-Gag antibody conjugated to a detection agent is added, resulting in a larger signal upon detection of Gag multimers (expected in late assembly intermediates) than upon detection of Gag monomers or dimers (expected in early assembly intermediates). Thus, small molecules that inhibit the assembly pathway are detected because they reduce the signal generated by the detection antibody. Under the theory that similar cellular machinery could be used by different viruses, analogous cell-free assembly screens were set up for seven other viruses that are human pathogens, and a master hit collection was generated consisting of 249 small molecules that inhibited assembly in one or more of these eight virus assembly screens [29].

Analysis of compounds from the master hit collection in cell culture assays identified a chemotype that was used for analog development, resulting in PAV-206, a small molecule that inhibits replication in an HIV-1 infected MT-2 T cell line and in HIV-1 infected human peripheral blood PBMCs with EC50s of 34 nM and 75 nM, respectively (Figure 4; [29]). Notably, studies of a PAV-206 analog in a HIV-1-infected human MT-2 T cells showed that this chemotype likely acts by inhibiting formation of the last assembly intermediate in the HIV-1 capsid assembly pathway [29].

## 6. Evidence That PAV-206 Acts by Targeting Host Complexes Critical for HIV-1 Assembly

The finding that PAV-206 analogs likely inhibit assembly of the immature HIV-1 capsid raised the possibility that these analogs bind directly to assembling Gag or to a host protein associated with assembling Gag, such as a host component of the Gag-containing assembly intermediates. Analysis of a biotinylated analog of PAV-206 using the proximity ligation assay (PLA) found that the PAV-206 analog displays a dose-dependent colocalization with Gag in HIV-1-expressing cells, while a biotinylated control compound that lacks antiretroviral activity does not colocalize with Gag (Figure 5A; [29]). While PLA is a highly sensitive method for demonstrating colocalization [79], finding the compound colocalized with Gag does not distinguish between the compound binding directly to Gag vs. binding to a host protein in a Gag-containing multiprotein complex.

A sensitive approach to determining whether a drug binds to a viral protein involves selecting for HIV-1 variants that are resistant to the drug. Every CA-binding inhibitor that has been studied extensively to date results in rapid development of resistant variants upon selection in vitro [22], including the two highly potent but broadly acting small molecules GS-CA1 and GS-6027 [77,80]. Similarly, the first and second generation maturation inhibitors, a different class of small molecules that inhibit Gag cleavage, also select for resistance mutations in vitro (reviewed in [81]).

Interestingly, upon selection for resistance to the immature capsid assembly inhibitor PAV-206 in MT-2 cells infected with HIV-1, no PAV-206-specific resistance mutations were identified during the 37 weeks of selection nor did high level resistance (at least 16-fold above the EC50) develop during that time [29]. This contrasted markedly with a selection experiment that was carried out in parallel using the well-studied protease inhibitor nelfinavir [82]; during the 37-week selection period, this control selection yielded three well known nelfinavir resistance mutations as well as resistance 16-fold above the EC50 [29]. The failure to develop rapid PAV-206-specific resistance upon selection suggested that PAV-206 does not bind directly to Gag or another viral protein. The lack of drug-specific resistance mutations, along with the finding that PAV-206 colocalizes with Gag, suggests that PAV-206 binds to a host factor associated with assembling Gag. Indeed, consistent with that possibility, experiments demonstrated that in addition to colocalizing with Gag, PAV-206 also colocalizes with two host factors that are associated with Gag in assembly intermediates—the two host enzymes that are known to facilitate assembly and reside in RNA-granule-like assembly intermediates, ABCE1 and DDX6 [29] (Figure 5A). Notably, controls showed that two other host proteins, G3BP1 found in stress granules and XRN1 found in P-bodies, are not colocalized with PAV-206. Since stress granules and P-bodies are prominent RNA granules that are not thought to play a role in HIV-1 assembly [29] (Figure 5B), these findings argue that the colocalization of PAV-206 with ABCE1 and DDX6 is not simply due to non-specific sticking of PAV-206 to host RNA granule proteins. Together the data support a model (Figure 6) in which PAV-206 likely binds to a host factor in the HIV-1 capsid assembly intermediates that are formed when Gag co-opts a host multiprotein complex that contains ABCE1, DDX6, and other host proteins. Association of PAV-206 with these RNA-granule-like intermediates appears to inhibit assembly of Gag, thereby reducing the production of viral particles.

The direct-binding target of PAV-206 and its analogs has not yet been identified, and much remains to be done before an analog of PAV-206 can be advanced to a preclinical stage. Nevertheless, there are important lessons in the discovery of this potent and specific small molecule inhibitor of Gag assembly. First, the findings show that a screen recapitulating the host-catalyzed HIV-1 assembly pathway led to the first potent and selective inhibitor of Gag assembly, after 15 years of screens based on CA and Gag self-assembly, indicating that screens that more closely recapitulate the cell biology of HIV-1 assembly can identify molecules missed by other screens. Second, the conclusion that the PAV-206 appears to target a host protein argues that host-targeting inhibitors of HIV-1 assembly could be highly successful. Third, identification of PAV-206 suggests that development of ARV drugs that will not select for resistant viral variants is an achievable goal—although it may require a shift in the paradigm that is used for target identification and antiviral drug development as discussed below. Finally, the success of this screen in discovering a potent assembly inhibitor is an important validation of the host-catalyzed model of HIV-1 capsid assembly model, since an inhibitor of this pathway would only be expected to block virus production if the pathway is indeed critical for virus assembly as has been proposed [41,45,46,49]. Thus, the finding that the first potent and specific inhibitor of assembly appears to be directed against a component of assembly intermediates is strong evidence that these complexes play a critical role in HIV-1 immature capsid assembly.

## 7. Host-Targeted Assembly Inhibitors—Challenges and Future Directions

Antiviral drugs that bind to viral proteins constitute the low-hanging fruit of antiviral drug development, attractive on many fronts. However, the best viral targets have been the subject of numerous screens over the past three decades and could be nearly fully tapped from a small molecule perspective. This leaves host-targeted therapies as the approach that is most likely to yield big gains in the future. While more complex and requiring novel approaches, host-targeting drugs are likely to yield the exciting reward of high genetic barriers that will greatly extend drug utility [83]. The identification of RABV and HIV assembly inhibitors through the use of screens that recapitulate host-catalyzed assembly pathways [29,78] makes a strong case that the main reason we lack host-targeting inhibitors of virus assembly is because we have not sufficiently understood host-catalyzed assembly pathways and explored these novel screening approaches.

Given the increasing dependence on ARV drugs not just for treatment but also for prevention [84,85], the importance of developing drugs that are less prone to development of resistance cannot be overstated. However, to achieve such success will require entirely new approaches. Screens aimed at identifying drugs that bind to host dependency factors, proteins known to facilitate virus events in the viral life cycle, may not be successful if such proteins are studied in isolation. Many viral–host interactions involve transient, poorly understood multiprotein complexes that likely contain host proteins that are “moonlighting”, i.e., performing a second non-canonical and often very different function, when present in a different context [86]. Screens that remove such cellular proteins from their multiprotein complex environment may be much less successful in identifying inhibitors of host-dependent steps of the viral life cycle than more holistic pathway screens. Such pathway screens, exemplified by the screens used to identify the HIV-1 and RABV assembly inhibitors described above, recapitulate a spectrum of key steps in virus assembly including translation of the capsid protein and the stepwise formation of assembly intermediates catalyzed by host enzymes [29,78].

Assembly pathway screens have two added advantages. First, they might identify small molecules that inhibit replication of unrelated viruses if those viruses use common host machinery during assembly [15]. Secondly, the multiprotein complexes involved in virus assembly could play important roles in non-viral diseases that result from disordered protein homeostasis [87,88], thereby leading to small molecule therapeutics effective against chronic degenerative diseases. For example, studies suggest that the neuronal protein ARC, which is found in synapses and is critical for learning and memory, evolved from an ancient retroelement Gag protein [89,90,91]; thus, the host machinery involved in Gag assembly could easily be critical for assembly of multiprotein complexes that are critical for neuronal function. Hence the importance of further understanding the host-targeting assembly inhibitor PAV-206. Diseases that result from disordered homeostasis and aggregation of neuronal multiprotein complexes include degenerative neurologic diseases for which few drug treatments exist [88]. Thus, from many perspectives, a better understanding of host-catalyzed assembly pathways and screens that result from them offers new and exciting avenues for combatting disease.

## Figures and Tables

**Figure 1 viruses-13-00451-f001:**
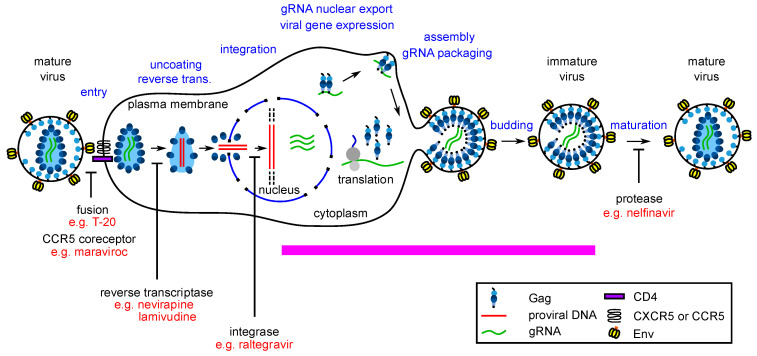
Diagram of the HIV-1 life cycle showing points of inhibition by FDA-approved drugs. Stages of the life cycle are labeled in blue font above the diagram. Events that are inhibited are shown with vertical blockade arrows below and labeled in black font, with an example of one drug in each category labeled in red font. The pink bar indicates late events in the viral life cycle that are not yet inhibited by an FDA approved drug, leaving a large gap in the ARV armamentarium.

**Figure 3 viruses-13-00451-f003:**
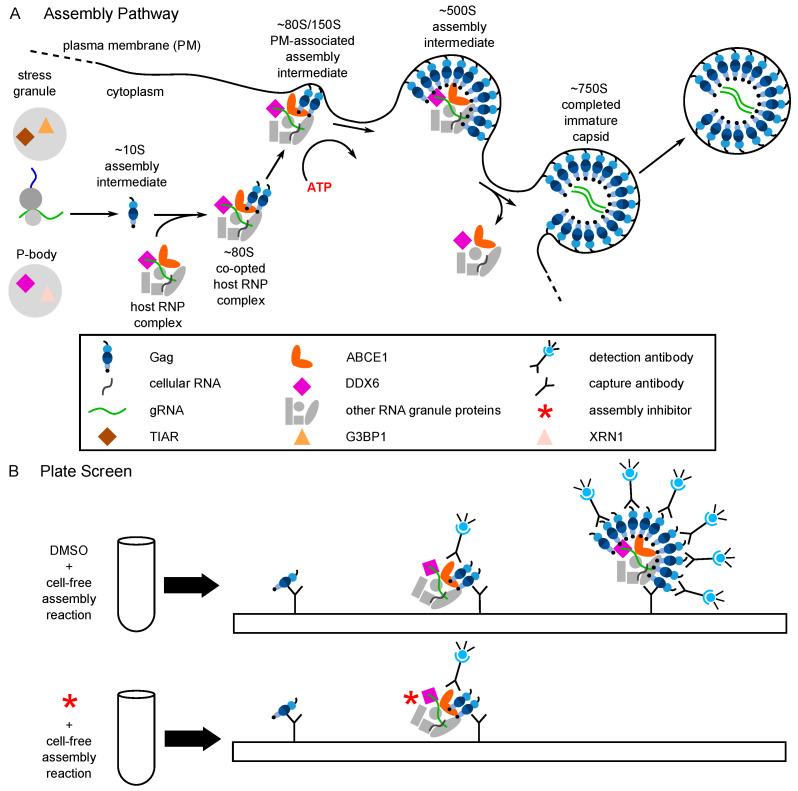
The host-catalyzed HIV-1 assembly pathway was developed into a screen for small molecule inhibitors. (**A**) Schematic showing the host-catalyzed HIV-1 assembly pathway, starting with Gag synthesis and formation of the early ~10S assembly intermediate. The ~80S assembly intermediate is formed when ~10S Gag co-opts a host RNP complex containing ABCE1 and DDX6, two host enzymes that have been shown to facilitate assembly. The co-opted host RNP complex is distinct from but related to P-bodies and stress granules, which are larger host RNP complexes. The ~80S assembly intermediate targets to the plasma membrane where Gag multimerization continues, resulting in sequential formation of the ~150S and ~500S assembly intermediates and the fully assembled ~750S immature capsid. Upon completion of assembly, the host RNP complex is released. Relevant references are in the text. (**B**) A cell-free protein synthesis and assembly plate screen that recapitulates the host-catalyzed HIV-1 assembly pathway was developed and utilized to identify small molecule inhibitors of the pathway [29]. In this screen, a capture antibody directed against Gag binds Gag monomers, oligomers, and multimers generated in a cell-free assembly reaction. The same anti-Gag antibody is used as a detection antibody that binds to captured oligomers and multimers, but not monomers whose binding site is occupied. The signal produced by the detection antibody is proportional to the amount of anti-Gag binding with a larger fluorescent signal indicating more extensive multimerization. The upper diagram shows the large signal that is produced when the HIV-1 cell-free assembly reaction is carried out in the presence of DMSO, which does not inhibit Gag assembly. The lower diagram shows inhibition of that signal when an inhibitor of Gag assembly is added at the start of the cell-free reaction. Legend in the middle applies to panels (**A**) and (**B**).

**Figure 4 viruses-13-00451-f004:**
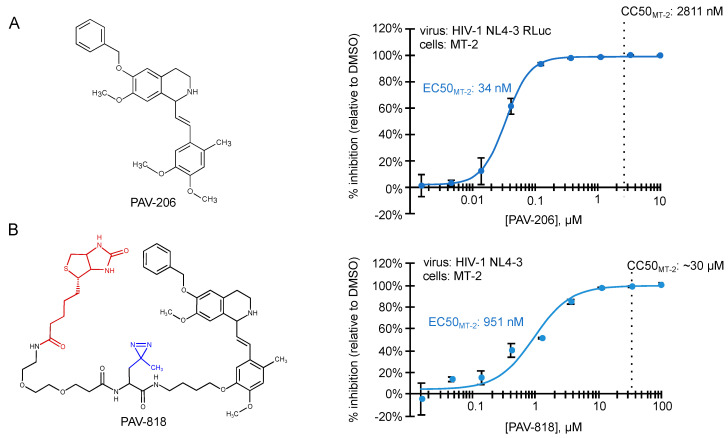
Small molecules that potently inhibit HIV-1 immature capsid assembly were recently described [29]. Structures of two ARV compounds are shown to the left. Dose–response curves for their inhibition of HIV-1 replication in the MT-2 T cell line are shown to the right, with EC50 and CC50 indicated. (**A**) PAV-206, is a tetrahydroisoquinolone derivative with excellent drug-like properties and a selectivity index (CC50/EC50) of ~82 in MT-2 cells. Other studies show that PAV-206 has a selectivity index of 48 in PBMCs and inhibits virus production most likely by reducing formation of the late ~500S assembly intermediate [29]. (**B**) PAV-818 is an analog of PAV-206 that retains ARV activity and contains the following modifications—a biotin tag for colocalization studies (shown in red) and a diazirine group for crosslinking (shown in blue). PAV-818 has a selectivity index (CC50/EC50) of ~32 in MT-2 cells.

**Figure 5 viruses-13-00451-f005:**
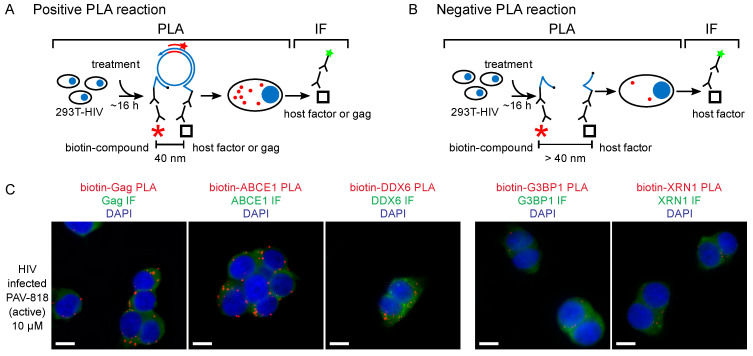
The biotinylated antiretroviral analog of PAV-206 colocalizes with three components of assembly intermediates but does not colocalize with two host proteins that are not associated with Gag or assembly intermediates [29]. (**A**) Schematic of the PLA approach for detecting colocalization of PAV-206 with either Gag or the host proteins ABCE1 and DDX6. 293T cells chronically infected with HIV-1 (293T-HIV) were treated with 10 µM PAV-818 (the biotinylated active compound). PLA was performed on treated cells by incubating with primary antibody pairs (either rabbit anti-biotin with mouse anti-Gag; mouse anti-biotin with rabbit anti-ABCE1; or mouse anti-biotin with rabbit anti-ABCE1) followed by incubation with PLA secondary antibodies (anti-rabbit IgG coupled to [+] PLA oligonucleotide and anti-mouse IgG coupled to [−] PLA oligonucleotide). Addition of other PLA reagents results in connector oligonucleotides annealing to the “+” and “–” oligonucleotides only if the primary antibodies are colocalized (within 40 nm); this in turn leads to the PLA amplification reaction. The addition of an oligonucleotide that recognizes a sequence in the amplified regions and is coupled to a red fluorophore (red star) results in intense spots only at sites where the two antibody targets (biotinylated compound and Gag, or biotinylated compound and host protein) are colocalized in situ. After PLA, IF was performed by adding secondary antibody conjugated to a green fluorophore (green star) to detect any unoccupied Gag or host protein antibody, thus marking Gag- or host-protein expressing cells with low-level green fluorescence. (**B**) This schematic illustrates a scenario in which the biotinylated compound and host protein are more than 40 nm apart. The connector oligonucleotide will not anneal to the antibody-conjugated [+] and [−] PLA oligonucleotides; thus, little to no amplification will occur and few or no red spots will be observed. (**C**) Data in the three panels on the left show colocalization by PLA of Gag, ABCE1, and DDX6 with PAV-818, respectively. Data in the two panels on the right show lack of colocalization with two host proteins that are found in P-bodies and stress granules, respectively, G3BP1 and XRN1, but are not known to be associated with Gag or found in assembly intermediates. Additional PLA negative controls, PLA dose–response curves, and quantitation are found in [29].

**Figure 6 viruses-13-00451-f006:**
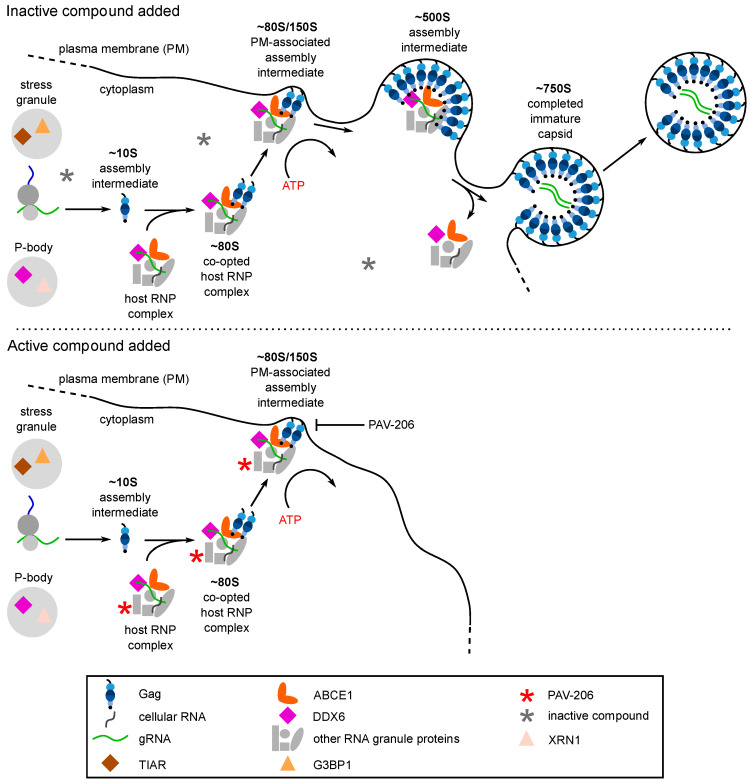
PAV-206 appears to inhibit HIV-1 replication by reducing formation of the ~500S assembly intermediate, most likely by binding to a host protein in assembly intermediates. The top panel shows that the assembly pathway is unaffected by an inactive control compound. For details of events in this pathway, see the Figure 3A legend. The bottom panel is based on recent findings [29] and shows that addition of PAV-206 or one of its active analogs results in association of the small molecule (red asterisk) with an as yet unidentified host protein in the assembly intermediates, leading to a reduction in the formation of the final ~500S assembly intermediate and little to no virus production.

## Data Availability

The data presented in this study are available within the article.

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
