# Peer review of "Addressing Antiretroviral Drug Resistance with Host-Targeting Drugs—First Steps towards Developing a Host-Targeting HIV-1 Assembly Inhibitor"

_viruses, 2021, doi:10.3390/v13030451_

Round 1

Reviewer 1 Report

The slowly increasing percentage of individuals failing current ART regiments in middle- and low-income countries has risen, presumably due to poor ART adherence and recently the COVID pandemic, and is a public health concern (UNAIDS report on the global AIDS epidemic 2020). The need for longer-lasting ART and newer drug treatments will be needed to maintain HIV suppression. Next generation capsid, RT, and integrase inhibitors are currently being evaluated and modifications to current ART to enhance in situ longevity may reduce ART adherence concerns. However, next generation small molecule HIV treatments targeting novel HIV sites or HIV host factors, and that are less prone to viral resistance, remain needed.  

 Lingappa and colleagues in their well-written submitted review provide a cursory overview of the treatment advantages of targeting HIV host factors, but focus mainly on reviewing just their published studies (JV, 2021, PMID: 33148797) on development and use of an in vitro capsid assembly assay to screen for small molecule inhibitors of host factors presumably required for cellular assembly / egress. The small molecule hits identified were found to reduce HIV production in cellular infection assays and the authors posit that the small molecules identified may be targeting host factors.

Overall, the authors follow the main discussion points of their just published JV article, which is of general interest to the field. However, the review does not provide additional insights beyond what was published in the JV article. Listed, below, are comments for the authors to consider addressing for the review article.

  • Please discuss the potential downsides of targeting host factors.
  • If I understand the authors correctly, they argue that compounds that disrupt virion assembly / egress have not yet been identified (passage 87-92). It seems that GS-CA1 (Nat Med. 2019 Sep;25(9):1377-1384, PMID: 31501601) appears to disrupt virion production.  The observation and publication should be mentioned.
  • The authors posit that the in vitro assay used may mimic their published model for viral assembly in the cell. While searching and reading the literature on the author’s extensive work modeling cellular HIV assembly, a recent publication on virion assembly suggests that the cellular pathway for virion assembly the authors proposed may open for interpretation (J Mol Biol. 2021 Feb, PMID: 33539875). Do these recent findings suggest a re-interpretation of the cellular mode of small molecule interaction(s) proposed by the authors?
  • Figure 3B. How do the authors know that they are capturing “assembly intermediates” rather than aggregated Gag multimers in their assays? Perhaps a little more discussion in the report on the assay would be helpful for the average reader.

Author Response

Response to Reviewer 1

We thank the reviewer for the thoughtful comments.  The reviewer described our manuscript as “well written” and “of general interest to the field”.  Listed below are changes we have made in response to each of the reviewer’s recommendations.  Line numbers refer to the revision of the manuscript that shows tracked changes. 

  1. The reviewer asked us to discuss the potential downsides of targeting host factors. This resulted in a new paragraph from lines 91 – 118.
  1. The reviewer asked us to mention a publication about GS-CA1 (Nat Med. 2019 Sep;25(9):1377-1384, PMID: 31501601), which appears to disrupt virion production. This publication (Yant et al., 2019, reference #59 in the original manuscript and # 77 in the revised manuscript) had been described (in lines 300-310 in the original submission).  In the original submission, we pointed out that this compound acts both early and late – and acts more potently on early events.   We have made edits emphasizing that although GS-CA1 (aka GS-6207) inhibits virus production, it does not do so selectively, and thus leaves us still lacking an assembly inhibitor that is as selective as the approved drugs that inhibit reverse transcription, entry, and integration (lines 426-429 in the revised manuscript).
  1. The reviewer asked us to comment on a new publication by Deng et al. J Mol Biol. 2021 Feb, PMID: 33539875). A preprint of this appeared in J Mol Biol after submission of our manuscript (with a final version published less than a week before this revised manuscript was submitted); hence why this paper was not discussed in our original submission.  We now answer the reviewer’s question about whether this paper suggests a reinterpretation of our findings (lines 302-367 of the revised manuscript).  While this section is a bit long, shortening it further would not allow us to explain to a general reader why we think the new study does not alter the interpretation of our findings. 
  2. The reviewer asked us how we know that we are capturing “assembly intermediates” rather than aggregated Gag multimers. We have now edited the section that explained that to improve clarity (lines 222-229 of the revised manuscript).  In addition, we comment further on the experiments that showed a precursor product relationship in our response to the paper by Deng et al. (lines 339-351 in the revised manuscript) and from a different perspective near that end of the manuscript (lines 586-590 in the revised manuscript). 

We hope the reviewer agrees that this revision addresses all the concerns and is now ready for publication. 

Reviewer 2 Report

In this review, the authors have gathered information about overcoming HIV-1 drug resistance via the identification of novel host-targeting antiretroviral drugs. This is an important and interesting review and will be a good source of information for a wide range of scientists including virologists, however, there are a few points that need to be addressed by the authors:

1-It could be very informative if the authors could briefly discuss the reasons/mechanisms for the development of resistance to ARV therapies in the introduction.

2-Although maraviroc has been shown to be safe and tolerable, it is important to mention the possible side-effects of host-targeting antiretroviral compounds.

3-Human genes should be italicized throughout the manuscript.

4-The focus of the review is about host-targeting antiretroviral strategies, however, there is still quite a lot of information about the ARV small molecules, e.g., page 2 which makes the current review a bit lengthy. This could be shortened to make the review more concise and focused.

5-A table summarizing different host-targeting strategies will be very helpful for the readership.

6-I suggest the authors put more emphasis on the benefits and disadvantages of host-targeting strategies compared to the ARV therapies, any idea to have a combination of both the therapies for a more effective therapeutic strategy could also be mentioned.

Author Response

Response to Reviewer 2

We thank Reviewer 2 for the useful and thoughtful feedback and comments.  The reviewer described our manuscript as “an important and interesting review [that] will be a good source of information for a wide range of scientists including virologists”.  Listed below are changes we have made in response to each of the reviewer’s recommendations.  Line numbers refer to the revision of the manuscript that shows tracked changes. 

  1. The reviewers asked us to briefly discuss the reasons/mechanisms for the development of resistance to ARV therapies in the introduction. We have now added this discussion (lines 58 – 71 of the revised manuscript). 
  2. The reviewer noted that it is important to mention the possible side-effects of host-targeting antiretroviral compounds. We agree that we were remiss in not addressing this directly.  We have now added paragraphs that discuss the disadvantages of host-targeting approaches (lines 93 – 123 of the revised manuscript).
  3. We agree with the reviewer that human genes should be italicized throughout the manuscript, but we only found names of human proteins, not genes, in our manuscript. Human proteins are not typically italicized; however, we will be in touch with the editors to make sure our manuscript complies with their standard formatting rules on this issue.
  4. The reviewer suggested that we reduce the amount of information about approved ARV small molecules, e.g., on page 2. However, this is difficult to do given the importance of emphasizing that only one of the 20 approved ARVs is a host-targeting agent.  Instead, we have edited this section to contextualize the approved ARVs as directing-acting vs. host-acting antiviral agents.  We hope this will serve to better integrate this section with the goal of the manuscript. 
  5. We were a bit puzzled by the reviewer’s request that we include a table “summarizing different host-targeting strategies”. Given that maraviroc is the only approved host-targeting antiretroviral agent, we don’t think a table is needed. The reviewer may have meant that such a summary should include all host-targeting agents that are under investigation and not yet approved.  However, including agents under investigation would greatly expand the length of this already fairly long manuscript. 
  6. The reviewers suggested that we put more emphasis on the advantages and disadvantages of host-targeting strategies and mention that combination of both host-targeting and direct-acting agents would be an effective therapeutic strategy. Advantages of host-targeting strategies were covered in the original version and have been edited for clarity (lines 72 – 91 of the revised manuscript). Disadvantages are covered in two paragraphs that we added in response to this comment and a similar one from Reviewer 1 (lines 93 – 123 of the revised manuscript).

We hope the reviewer agrees that this revision addresses all the concerns and is now ready for publication.